

# Predictive values of stool-based tests for mucosal healing among Taiwanese patients with ulcerative colitis: a retrospective cohort analysis

Hsu-Heng Yen[1,2,3,*], Mei-Wen Chen[4,5,*], Yu-Yao Chang[6], Hsuan-Yuan Huang[6], Tsui-Chun Hsu[1] and Yang-Yuan Chen[1]

[1] Division of Gastroenterology, Department of Internal Medicine, Changhua christian Hospital, Changhua, Taiwan
[2] Institute of Medicine, Chung Shan Medical and Dental College, Taichung, Taiwan
[3] General Education Center, Chienkuo Technology University, Changhua, Taiwan
[4] Department of Tumor Center, Changhua Christian Hospital, Changhua, Taiwan
[5] Information Management, Chienkuo Technology University, Changhua, Taiwan
[6] Department of Colorectal Surgery, Changhua Christian Hospital, Changhua, Taiwan
[*] These authors contributed equally to this work.

Corresponding author
Yu-Yao Chang, 177176@cch.org.tw

## ABSTRACT

**Background/Purpose.** Over the past two decades, ulcerative colitis (UC) has emerged in the Asia Pacific area, and its treatment goal has shifted from symptom relief to endoscopic remission. Endoscopy is the gold standard for the assessment of mucosal healing; however, it is an invasive method. Fecal calprotectin (FC) is a non-invasive stool-based inflammatory marker which has been used to monitor mucosal healing status, but it is expensive. By contrast, the immune fecal occult blood test (iFOBT) is a widely utilized stool-based screening tool for colorectal cancer. In this study, we compared the predictive values of iFOBT and FC for mucosal healing in Taiwanese patients with UC.

**Methods.** A total of 50 patients with UC identified via the electronic clinical database of Changhua Christian Hospital, Taiwan, were retrospectively enrolled from January 2018 to July 2019. Results of iFOBT, FC level, and blood tests as well as Mayo scores were reviewed and analyzed. Colonic mucosa was evaluated using the endoscopic Mayo subscore.

**Results.** The average age of the patients was 46 years, and 62% of the patients were men. Disease distribution was as follows: E1 (26%), E2 (40%), and E3 (34%). Complete mucosal healing (Mayo score = 0) was observed in 30% of patients. Endoscopic mucosal healing with a Mayo score of 0 or 1 was observed in 62% of the patients. Results of FC and iFOBT were compared among patients with and without mucosal healing. Predictive cutoff values were analyzed using receiver operating characteristics curves. iFOBT and FC had similar area under the curve for both complete mucosal healing (0.813 *vs.* 0.769, respectively, $p = 0.5581$) and endoscopic mucosal healing (0.906 *vs.* 0.812, respectively, $p = 0.1207$).

**Conclusion.** In daily clinical practice, FC and iFOBT do not differ in terms of predictive values for mucosal healing among Taiwanese patients with UC.

## INTRODUCTION

Ulcerative colitis (UC) is a chronic inflammatory disease of the colon (*Wei et al., 2017*; *Yen et al., 2017*; *Yen et al., 2019*); without adequate treatment and monitoring of the disease, it may lead to complications such as bleeding, perforation, and development of malignancy (*Wei et al., 2017*). The state-of-the-art treatment goal for UC has shifted from clinical remission with symptom control to endoscopic remission using the treat-to-target strategy (*Jackson & De Cruz, 2019*; *Rubin et al., 2019*; *Wei et al., 2017*). Fecal calprotectin (FC) is a non-invasive fecal marker commonly used in Western countries to determine mucosal healing (*Freeman et al., 2019*; *Motaganahalli et al., 2019*); however, FC is expensive compared with the immune fecal occult blood test (iFOBT) and is not reimbursed for clinical use in Taiwan. By contrast, colonoscopy remains the gold standard for the assessment of colonic mucosal status, enables screening for colitis-associated malignancies (*Wei et al., 2017*; *Yen et al., 2017*), and costs less in Asia than in Western countries (*Chang et al., 2020*; *Yen & Hsu, 2019*). However, colonoscopy is invasive and patients may show slight reluctance to undergo this procedure (*Lin et al., 2015*; *Yen & Hsu, 2019*). Moreover, colonoscopy may induce worsening of UC or result in complications (*Wei et al., 2017*).

Quantitative iFOBT has replaced guaiac-based measures of stool hemoglobin concentrations owing to its better performance in colorectal cancer screening. Recent studies conducted in Canada (*Ma et al., 2017*), Japan (*Nakarai et al., 2013*), Korea (*Ryu et al., 2016*), and China (*Shi et al., 2017*) used iFOBT to predict mucosal healing among patients with UC (*Dai et al., 2018*). Meanwhile, only limited data are available on the comparison of the relative predictive values of iFOBT and FC for mucosal healing (*Kim et al., 2020*; *Naganuma et al., 2020*; *Ryu et al., 2019*; *Takashima et al., 2015*). In 2018, with the aim of improving clinical practice, we began performing iFOBT and FC at our institution for disease monitoring of patients with inflammatory bowel disease (IBD). In the present study, we aimed to compare the predictive values of iFOBT and FC for mucosal healing in Taiwanese patients with UC.

## MATERIALS & METHODS

The medical records of patients diagnosed with UC between January 2018 and July 2019 at Changhua Christian Hospital, Taiwan, were retrospectively reviewed. From January 2018, patients diagnosed with IBD, including UC and Crohn's disease, received integrated hospital care by a trained IBD nurse, who recorded clinical symptoms using the Mayo scoring system for UC severity during each outpatient clinic visit. Patients with UC underwent screening colonoscopy for the evaluation of disease activity and monitoring of malignancy. Laboratory tests were conducted annually during the follow-up period. Starting from 2018, prior to colonoscopy, stool samples were collected for an immunochemical occult test to evaluate and compare the usefulness of iFOBT and FC for evaluating IBD activity in clinical practice. Samples for iFOBT were analyzed using the HM-JACK system

(Kyowa Medex, Shizuoka, Japan), which is a fully automated quantitative iFOBT system. The HM-JACK system can accurately measure fecal hemoglobin concentrations within a range of 7–400 ng/mL. Samples for FC were analyzed using the commercially available Quantum Blue® fCAL test (Buhlmann Laboratories AG, Schonenbuch, Switzerland).

Patients enrolled in the present study met the following inclusion criteria: (a) diagnosed with UC for > 6 months; (b) tested by both iFOBT and FC within 1 month prior to colonoscopy; (c) had medical records of clinical symptoms, partial Mayo score, and laboratory test results, and (d) had undergone colonoscopy with documentation of the endoscopic Mayo scoring system. The Mayo score and laboratory test data were obtained 1 month before colonoscopy.

The requirement for informed consent for data extraction was waived by the institutional review board because of the retrospective design of the study and the minimal risk involved. Mayo scores, including stool frequency, rectal bleeding, endoscopy findings, and overall clinical evaluation, were used to evaluate UC severity using 0–3 points for each component. The partial Mayo score comprised three non-endoscopic variables. An endoscopic Mayo score of 0 indicates complete mucosal healing (CMH), and an endoscopic Mayo score of 0 or 1 on colonoscopy indicates endoscopic mucosal healing (EMH). The primary endpoint of this study was to compare the predictive values of stool-based tests for the mucosal status, i.e., CMH and EMH. The secondary endpoint was to compare the correlation between endoscopy activity and stool-based tests and blood-based systematic inflammatory markers. The study complied with the World Medical Association Declaration of Helsinki for medical research involving human subjects including research on identifiable human material and data. The study was approved by the Institutional Review Board of Changhua Christian Hospital (approval number: CCH IRB 190814).

## Statistical analysis

The extracted data were organized using Microsoft Excel and analyzed using MedCalc Statistical Software version 19.16 (MedCalc Software bvba, Ostend, Belgium; https://www.medcalc.org; 2020). Continuous data are expressed as means and standard deviations or as medians and interquartile ranges for normally and non-normally distributed data, respectively. Categorical variables are presented as numbers and percentages. The mean values of normally distributed variables were compared by an independent sample's Student's $t$-test. Mann–Whitney $U$-test and Kruskal–Wallis test were performed to compare the mean values of 2 and $\geq 3$ groups of variables, respectively, with non-normal distributions. The frequencies of categorical variables were compared using Pearson's $\chi^2$ or Fisher's exact test, as appropriate. Spearman's rank correlation was performed to determine the correlation between fecal test data and UC severity as reflected by the Mayo score. Receiver operating characteristic (ROC) curve analysis was conducted to determine the best cutoff values of iFOBT and FC levels for predicting mucosal healing. All $P$-values of < 0.05 were considered statistically significant.

**Table 1  Demographic data of the patients.**

| Clinical variable | |
|---|---|
| Sex (M/F) | 31/19 |
| Age, years (mean, 95% CI) | 46 (39 ∼52.4) |
| Disease duration (median, IQR) | 3.5 (2–6) |
| Disease distribution (E1/E2/E3) | 13/20/17 (26%/40%/34%) |
| Smoking status (nonsmoker/current smoker/ever-smoker) | 34/4/12 (68%/8%/24%) |
| Current medication | |
| − steroid (Y/N) | 17/33 (34%/66%) |
| − oral 5-ASA (Y/N) | 47/3 (94%/6%) |
| − rectal 5-ASA (Y/N) | 27/23 (54%/47%) |
| − immune modulator (Y/N) | 12/38 (24%/76%) |
| - biologic agent (Y/N) | 1/49 (2%/98%) |
| WBC count, $10^3/\mu$L (median, IQR) | 6.5 (5.6–7.4) |
| Hemoglobin level, g/dL (median, IQR) | 14.15 (13.26–14.6) |
| Platelet count,$10^3/\mu$L (median, IQR) | 261 (229.83–295.95) |
| NLR (median, IQR) | 2.26 (1.84–2.74) |
| CRP level, mg/dL (median, IQR) | 0.13 (0.09 –0.24) |
| ESR, mm/h (median, IQR) | 10 (8–13.8) |
| Endoscopy Mayo score (0/1/2/3) | 15/16/8/11(30%/32%/16%/22%)) |
| iFOBT, ng/ml (median, IQR) | 44.5 (7–101.7) |
| FC, $\mu$g/g (median,IQR) | 135.88 (83.06–651.6) |

## RESULTS

### Clinical features of patients with UC

During the study period, a total of 102 patients with UC received treatment at the hospital and 50 met the inclusion criteria. The clinical characteristics of all patients are presented in Table 1. The average age of the patients was 46 years, and the majority of them were men (31 men, 19 women). The median duration of UC was 3.5 years. Based on the Montreal classification, the diseases observed included proctitis (E1, 26%), left-sided UC (E2, 40%), and extensive UC (E3, 34%). Further, there were 8% current smokers, 24% ever-smokers, and 68% non-smokers. Oral 5-aminosalicylic acid (5-ASA) was administered to 94% of the patients, followed by rectal 5-ASA (54%), steroids (34%), and immune modulators (24%). CMH with an endoscopic Mayo score of 0 was achieved in 30% of the patients, whereas EMH)with an endoscopic Mayo score of $\leq 1$ was achieved in 62%.

### Comparison of patients with and without mucosal healing

In our cohort, 30% (15/50) of the patients exhibited CMH as evaluated by colonoscopy; the remaining patients had endoscopic Mayo scores of 1 (33.3%), 2 (18.52%), and 3 (22.22%) (Table 2). Age, sex, disease distribution, and drugs used did not differ between patients with CMH and those without CMH. Compared with patients without CMH, those with CMH had lower iFOBT ($P = 0.003$) and FC ($P = 0.0028$) values.

EMH, as indicated by an endoscopic Mayo score of 0 or 1 on colonoscopy, was achieved in 65% (Table 3). Age, sex, disease distribution, and drugs used did not differ between

**Table 2** Comparison of patients with and without complete mucosal healing.

|  | CMH (+) (n = 15) | CMH (−) (n = 35) | P-value |
|---|---|---|---|
| Age (mean, SD) | 48.13 (15.17) | 45.71 (12.51) | 0.560 |
| WBC count (median, IQR) | 5.7 (5.4–8.6) | 6.7 (5.4–8.2) | 0.8489 |
| Hb (median, IQR) | 16.7 (13.8–15.3) | 17 (11.4–14.7) | 0.0284 |
| Platelet (median, IQR) | 233 (192–265) | 287 (220–350)) | 0.0466 |
| iFOBT, ng/mL (median, IQR) | 7 (7–22.5) | 121 (8.5–400) | 0.0003 |
| FC, μg/g (median, IQR) | 59.95 (12.88–110.5) | 555 (79.96–1687.34) | 0.0028 |
| Disease distribution (E1/E2/E3) | 3/7/5 | 10/13/12 | 0.7643 |
| Mayo score stool frequency (0/1/2/3) | 4/9/2/0 | 7/13/10/5 | 0.0694 |
| Mayo score rectal bleeding (0/1/2/3) | 12/3/0/0 | 17/8/8/2 | 0.0162 |
| Physician rating of disease activity (0/1/2/3) | 9/6/0/0 | 7/20/7/1 | 0.0032 |
| Current medication |  |  |  |
| –steroids (%) | 20% | 40% | 0.1756 |
| –oral 5-ASA (%) | 93.3% | 94.3% | 0.8976 |
| –rectal 5-ASA (%) | 60% | 51.4% | 0.5812 |
| –immune modulator (%) | 20% | 25.7% | 0.6678 |
| − biologic agent (%) | 6.7% | 0% | 0.1266 |

**Notes.**
5-ASA, 5-aminosalicylic acid; BMI, body mass index; CMH, complete mucosal healing; Hb, hemoglobin; iFOBT, immune fecal occult blood test; WBC, white blood cell.

those with EMH and those without EMH. Compared with patients without EMH, those with EMH had lower iFOBT ($P < 0.001$) and FC ($P = 0.0002$) values.

**Relative predictive values of iFOBT and FC for mucosal healing**

Both iFOBT and FC showed moderate correlations with the total endoscopic Mayo score (Fig. 1). The correlation among stool-based tests (FC and iFOBT), blood inflammatory markers (CRP and ESR), and endoscopically evaluated UC activity was analyzed using Spearman's rank correlation coefficient (Table 4). Compared with FC, ESR, and CRP, iFOBT exhibited a higher correlation with the endoscopic Mayo score. As illustrated in Fig. 2, an iFOBT criterion of ≤30 ng/mL had 93.33% sensitivity and 71.43% specificity in terms of predicting CMH, whereas an iFOBT criterion of ≤43 ng/mL had 80.65% sensitivity and 100% specificity in predicting EMH. As illustrated in Fig. 3, an FC criterion of ≤ 156 μg/g had 86.67% sensitivity and 62.86% specificity in terms of predicting CMH, whereas an FC criterion of ≤ 156 μg/g had 74.19% sensitivity and 84.21% specificity in predicting EMH. As illustrated in Fig. 4, the ROC curve analysis of the ability to predict mucosal healing showed that iFOBT results tended to have higher AUCs for both CMH and EMH than FC values.

## DISCUSSION

In the present study, we compared the predictive values of iFOBT and FC for mucosal healing among Taiwanese patients with UC in clinical practice. Our findings were consistent with those of previous studies, which demonstrated that iFOBT and FC had similar predictive values for predicting EMH among patients with UC (*Kim et al., 2018*;

**Table 3** Comparison of patients with and without endoscopic mucosal healing.

|  | EMH (+) ( $n = 31$) | EMH (−) ( $n = 19$) | *P*-value |
|---|---|---|---|
| Age (mean, SD) | 47.16 (15.26) | 45.26 (9.38) | 0.628 |
| WBC count (median, IQR) | 6.1 (5.1–7.7) | 7.4 (5.5–8.7) | 0.2630 |
| Hb (median, IQR) | 14.2 (13.4–15.1) | 12.7 (9.9–14.7) | 0.0324 |
| Platelet (median, IQR) | 246 (207–281) | 319 (249–403) | 0.0115 |
| iFOBT, ng/mL (median, IQR) | 7 (7–29.5) | 283 (129.3–400) | <0.0001 |
| FC, μg/g (median, IQR) | 71.2 (39.1–222.8) | 912 (448.8–1800) | 0.0002 |
| Disease distribution (E1/E2/E3) | 8/14/9 | 5/6/8 | 0.5665 |
| Mayo score stool frequency (0/1/2/3) | 11/16/3/1 | 0/6/9/4 | 0.0001 |
| Mayo score rectal bleeding (0/1/2/3) | 23/6/2/0 | 6/5/6/2 | 0.0079 |
| Physician rating of disease activity (0/1/2/3) | 15/15/1/0 | 1/11/6/1 | 0.0015 |
| Current medication |  |  |  |
| –steroids (%) | 22.6% | 52.6% | 0.0311 |
| –oral 5-ASA (%) | 93.55 | 94.7% | 0.8650 |
| –rectal 5-ASA (%) | 54.8% | 52.6% | 0.8804 |
| –immune modulator (%) | 19.4% | 31.6% | 0.3308 |
| − biologic agent (%) | 3.2% | 0% | 0.4337 |

**Notes.**
5-ASA, 5-aminosalicylic acid; BMI, body mass index; CMH, complete mucosal healing; EMH, endoscopic mucosal healing; Hb, hemoglobin; iFOBT, immune fecal occult blood test; WBC, white blood cell.

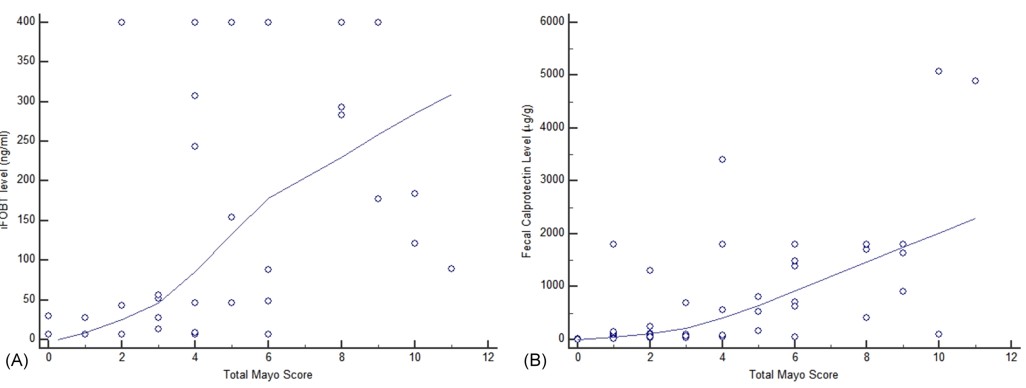

**Figure 1** **Correlation of complete Mayo score with iFOBT (A) and FC (B).** (A) Spearman's coefficient of rank correlation = 0.737, $P < 0.0001$ (B) Correlation of FC with a complete Mayo Score.

*Kim et al., 2020*; *Ma et al., 2017*; *Naganuma et al., 2020*; *Ryu et al., 2019*; *Takashima et al., 2015*). iFOBT might be used as an alternative non-invasive monitoring tool for patients with UC, particularly in Taiwan, where other fecal tests such as FC level are not widely available and are more expensive.

The incidence of IBD is increasing worldwide, particularly in the Asia Pacific region (*Jung, 2020*; *Wei et al., 2013*; *Yen et al., 2019*). Our recent national cohort study from Taiwan (*Yen et al., 2019*) reported a six-fold increase to 12.8/100,000 in the prevalence of UC over the past 15 years. The treatment goal for IBD has shifted from clinical remission to biochemical

**Table 4  Correlations of endoscopy activity with iFOBT, FC, ESR, and CRP.**

|  | Mayo Endoscopy Score | iFOBT | ESR | FC | CRP |
| --- | --- | --- | --- | --- | --- |
| Mayo Endoscopy Score |  | 0.708 | 0.449 | 0.548 | 0.497 |
| iFOBT | 0.708, $p < 0.0001$ |  | 0.481 | 0.568 | 0.316 |
| ESR | 0.449, $p = 0.0012$ | 0.481, $p = 0.0005$ |  | 0.312 | 0.607 |
| FC | 0.548, $p < 0.0001$ | 0.568, $p < 0.0001$ | 0.312, $p = 0.0291$ |  | 0.317 |
| CRP | 0.497, $p = 0.0003$ | 0.316, $p = 0.0271$ | 0.607, $p < 0.0001$ | 0.317, $p = 0.0266$ |  |

**Notes.**
Spearman rank correlation coefficient.
CRP, C-reactive protein; ESR, erythrocyte sedimentation rate; iFOBT, immune fecal occult blood test; FC, fecal calprotectin.

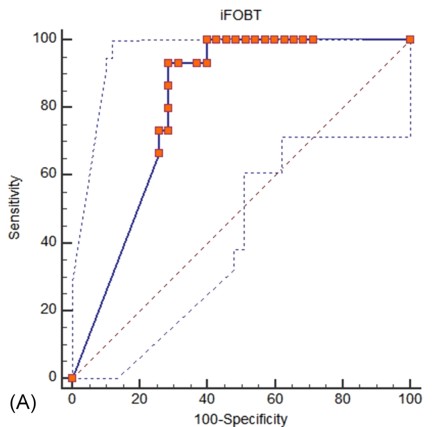
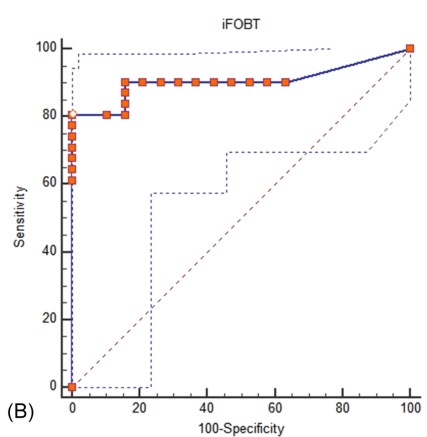

**Figure 2  ROC curve analysis for the use of iFOBT to assess complete mucosal healing (A) and endoscopic mucosal healing (B).** (A) Using an iFOBT criterion of $\leq 30$ ng/mL (95% CI $\leq 27$ to $\leq 52$), the sensitivity is 93.33%, and the specificity is 71.43% for predicting complete mucosal healing. Dot line: 95% Confidence interval of ROC curve. (B) Using an iFOBT criterion of $\leq 43$ ng/mL (95% CI $\leq 28$ to $\leq 88$), the sensitivity is 80.65%, and the specificity is 100% for predicting endoscopic mucosal healing. Dot line: 95% Confidence interval of ROC curve.

remission, endoscopic remission, and histological healing (*Rubin et al., 2019*; *Wei et al., 2017*). Patients with UC are typically evaluated using clinical symptoms based on the Mayo scoring system. However, subjective reports of symptoms such as bowel frequency may not correlate well with the endoscopy findings. Uncontrolled but asymptomatic inflammation may increase the risk of disease relapse or subsequent development of complications (*Peyrin-Biroulet et al., 2015*). Colonoscopy has been the gold standard for the evaluation of mucosal status; however, it is invasive and less acceptable by patients compared with non-invasive tests such as blood and stool tests. In the present study, 25% of the patients with an endoscopic Mayo score of 3 for severe colonic inflammation reported no rectal bleeding and/or had normal or near-normal stool frequencies. Therefore, relying on patient-reported symptoms may underestimate the severity of colonic inflammation.

Calprotectin is a calcium-binding protein, which is mainly found in neutrophils (*Wei, 2016*). FC has been correlated with mucosal inflammation and has been used as a surrogate biomarker for evaluating IBD activity in Western countries (*Kim et al., 2020*; *Lin et al.,*

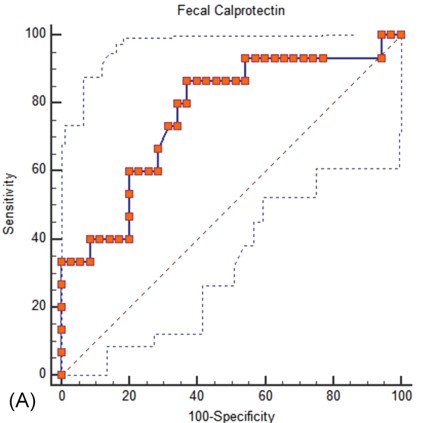
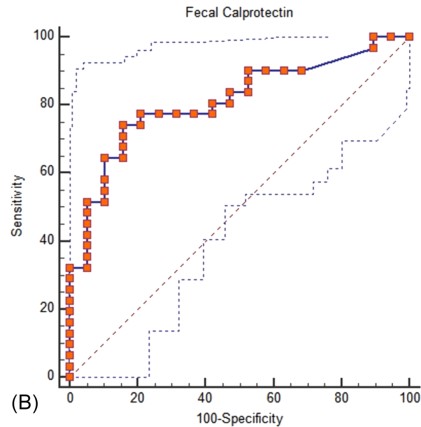

(A)      (B)

**Figure 3** **ROC curve analysis for the use of FC to assess complete mucosal healing (A) and endoscopic mucosal healing (B).** (A) Using an FC criterion of ≤156 μg/g (95% CI ≤100 to ≤1800), the sensitivity is 86.67%, and the specificity is 62.86% for predicting complete mucosal healing. Dot line: 95% Confidence interval of ROC curve. (B) Using an FC criterion of ≤156 μg/g (95% CI ≤63.7 to ≤638.9), the sensitivity is 74.19% and the specificity is 84.21% for predicting endoscopic mucosal healing.Dot line: 95% Confidence interval of ROC curve.

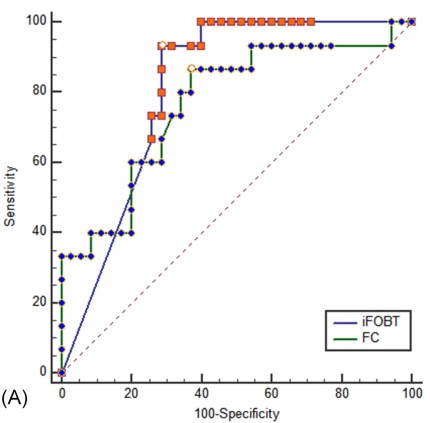
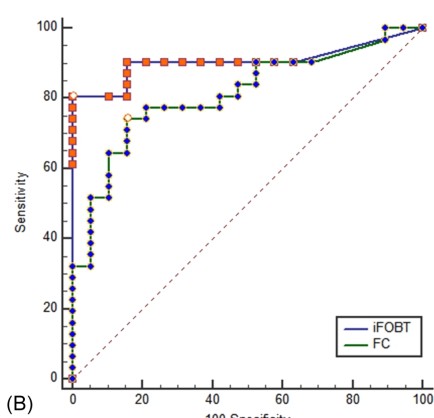

(A)      (B)

**Figure 4** **ROC curve comparing iFOBT and FC for predicting complete mucosal healing (A) and endoscopic mucosal healing (B).** (A) Pairwise comparison of ROC curves: iFOBT vs. FC, $p = 0.5581$. (B) Pairwise comparison of ROC curves: iFOBT vs. FC, $p = 0.1207$.

*2015*; *Takashima et al., 2015*). Although non-invasive, FC is not readily available in Asia and it costs the same as or more than colonoscopy in Taiwan (*Lin et al., 2015*; *Wei et al., 2017*); therefore, its use in our clinical practice has been limited. By contrast, quantitative iFOBT is a stool-based test for colon cancer screening and is available worldwide (*Yen & Hsu, 2019*). Unlike Crohn's disease, UC involves the superficial colonic wall; therefore, mucosal hemorrhage may be used as a surrogate marker for predicting mucosal damage (*Kuriyama et al., 2010*).

In a study from Canada, *Ma et al. (2017)* reported similar predictive values of iFOBT and FC for mucosal healing in patients with IBD, particularly with UC. Further, in Asia, the

interest in the use of iFOBT as a test for predicting mucosal damage among patients with UC has been increasing (*Naganuma et al., 2019*; *Nakarai et al., 2013*; *Ryu et al., 2016*; *Ryu et al., 2019*; *Shi et al., 2017*; *Takashima et al., 2015*). Using different cutoff levels and test kits, iFOBT was reported to predict endoscopy mucosal healing with 58%–94.9% sensitivity and 38.3%–100% specificity. Data on the optimal cutoff levels of these stool-based tests for predicting EMH has varied among different studies due to the different testing kits used.

Although the UC treatment guidelines in Taiwan have recommended FC as a biomarker for detecting colonic inflammation and for evaluating disease activity (*Wei et al., 2017*), this test is not widely available and is not reimbursed by the national insurance in Taiwan. Few studies have compared the performances of iFOBT and FC (Table 5), and the results are conflicting. *Takashima et al. (2015)* conducted the first study on the use of FC and iFOBT to predict the mucosal status of 92 patients with UC. They found that both iFOBT and FC efficiently predicted mucosal healing in UC, but compared with FC, iFOBT appeared to be more sensitive in predicting CMH. *Ryu et al. (2019)* from Korea found a more accurate prediction of endoscopic activity with FC than with iFOBT. *Kim et al. (2018)*, also from Korea, found that compared with iFOBT, FC was better in predicting CMH but with a similar performance with regard to predicting EMH (*Kim et al., 2018*; *Kim et al., 2020*). To the best of our knowledge, the present study is the first from Taiwan to compare the relative predictive values of iFOBT and FC. Our findings are consistent with those of a recent Japanese nationwide cohort study (*Naganuma et al., 2020*), which suggested similar predictive values of iFOBT and FC for the mucosal status of patients with UC. Given the low cost and similar predictive value of FC, iFOBT may be advantageous as a first-line monitoring tool for patients with UC in Taiwan.

There are several limitations to this study. First, our study had a single-center retrospective design; therefore, the results require validation by further large-scale studies outside Taiwan. Endoscopic findings were retrospectively reviewed and may have been biased due to the fact that different endoscopists were involved. Second, we only performed one iFOBT and FC test before colonoscopy owing to the high cost of FC. Therefore, we were not able to investigate and compare the potential diagnostic relevance of iFOBT and FC in predicting treatment response or disease relapse after remission. The combined use of both fecal tests is promising, and Japanese researchers have shown promising results with regard to predicting disease relapse (*Naganuma et al., 2020*; *Nakarai et al., 2018*). Third, colonic polyps or cancer was not identified in any of our patients during colonoscopy. Theoretically, iFOBT cannot differentiate between UC and colorectal neoplasms. Importantly, iFOBT should not replace surveillance colonoscopy; we suggest its use as a monitoring tool to assess disease activity rather than for cancer surveillance.

## CONCLUSIONS

The increased prevalence of UC in Taiwan has raised the need for a practical tool to monitor disease activity. Our study found similar predictive values of iFOBT and FC for both CMH and EMH. Therefore, iFOBT might be useful as a first-line non-invasive tool in clinical practice to evaluate disease severity and may assist in clinical decision making.

**Table 5  Literature comparing the use of FC and iFOBT for the prediction Mayo endoscopic mucosal healing among patients with UC.**

| Author, Region | Case Number, Study Type, Center | Description of Results | Year |
|---|---|---|---|
| Yen, Taiwan | 50, R, S | No difference in AUC of FC (cutoff level, 156 µg/g) vs. iFOBT (cutoff level, 30 ng/mL) for MES = 0 vs. MES ≥ 1 (0.769 vs. 0.813, $P = 0.5581$) No difference in AUC of FC (cutoff level, 156 µg/g) vs. iFOBT (cutoff level, 43 ng/mL) for MES ≤1 vs. MES ≥2 (0.812 vs. 0.906, $P = 0.1207$) | Present Study |
| Naganuma, Japan | 429, P,M | No difference in AUC of FC (cutoff level, 146.0 mg/kg) vs. iFOBT (cutoff level, 77.0 ng/mL) for MES = 0 vs. MES ≥ 1 (0.7774 vs. 0.8085, $P = 0.394$) No difference in AUC of FC (cutoff level, 277.0 mg/kg) vs. iFOBT (cutoff level, 201.0 ng/mL) for MES ≤ 1 vs. MES ≥2 (0.8166 vs. 0.8353, $P = 0.394$) | 2020[17] |
| Kim, Korea | 127, P, M | AUC of FC (cutoff level, 70 µg/g) >iFOBT (cutoff level, 0 ng/mL) for MES = 0 vs. MES ≥ 1 (0.858 vs. 0.707, $P = 0.0009$) No difference in AUC of FC (cutoff level, 200 µg/g) vs. iFOBT (cutoff level, 60 ng/mL) for MES ≤1 vs. MES ≥2 (0.82 vs. 0.813, $P = 0.089$) | 2020[16] |
| Ryu, Korea | 128, P, S | AUC of FC (cutoff level, 170 µg/g) >iFOBT (cut-off value,100 ng/mL )for MES = 0 vs. MES ≥ 1 (0.847 vs. 0.757, $P < 0.0001$) AUC of FC (cutoff level, 170 µg/g)>IFOBT (cut-off value,100 ng/mL) for MES ≤1 vs. MES ≥ 2 (0.863 vs. 0.765, $P < 0.0001$) | 2019[18] |
| Kim, Korea | 68,R,S | No difference in AUC of FC vs. iFOBT for MES ≤ 1 vs. MES ≥2 (0.727 vs. 0.717, $P = 0.8643$) | 2018[20] |
| Takashima, Japan | 92,P,S | No difference in AUC of FC (cut-off value, 200 µg/g) vs. iFOBT (cut-off value, 75 ng/ml) for MES = 0 vs. MES ≥ 1 (0.83 vs. 0.82, $P = 0.394$) No difference in AUC of FC (cut-off value, 369 µ g/g) vs. iFOBT (cut-off value, 280 ng/ml) for MES ≤1 vs. MES ≥2 (0.80 vs. 0.79, $P = 0.394$) | 2015[19] |

Notes.
P, prospective; S, single-center study; M, multicenter study.

## Funding
This work was supported by Changhua Christian Hospital (108-CCH-IRP-018). The funders had no role in study design, data collection and analysis, decision to publish, or preparation of the manuscript.

## Grant Disclosures
The following grant information was disclosed by the authors:
Changhua Christian Hospital: 108-CCH-IRP-018.

## Competing Interests
The authors declare there are no competing interests.
## Author Contributions

- Hsu-Heng Yen and Yu-Yao Chang conceived and designed the experiments, performed the experiments, analyzed the data, prepared figures and/or tables, authored or reviewed drafts of the paper, and approved the final draft.
- Mei-Wen Chen performed the experiments, analyzed the data, authored or reviewed drafts of the paper, and approved the final draft.
- Hsuan-Yuan Huang and Tsui-Chun Hsu performed the experiments, authored or reviewed drafts of the paper, and approved the final draft.
- Yang-Yuan Chen conceived and designed the experiments, performed the experiments, prepared figures and/or tables, authored or reviewed drafts of the paper, and approved the final draft.

## Human Ethics

The following information was supplied relating to ethical approvals (i.e., approving body and any reference numbers):

Institutional Review Board of Changhua Christian Hospital approved the study (CCH IRB 190814).

## Data Availability

The raw measurements are available as a Supplemental File.

## Supplemental Information

Supplemental information for this article can be found online at http://dx.doi.org/10.7717/peerj.9537#supplemental-information.

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
