# Peer review of "Predictive values of stool-based tests for mucosal healing among Taiwanese patients with ulcerative colitis: a retrospective cohort analysis"

_PeerJ, doi:10.7717/peerj.9537_

## Round 0.1 · original submission · Major Revisions

The authors have to address the concerns raised by the reviewers.

Reviewer 1 ·

Basic reporting

1. The English language should be improved to ensure that an international audience can clearly understand your text.
2. The font size of the all figures should be identical.
3. Figure 2 was uploaded incompletely.

Experimental design

1. In part of Materials & Methods, more details should be provided. I am confused that “the correlation of iFOBT and FC with the total Mayo score…” whether the total Mayo score was the baseline score. Additionally the evaluation points of the EMH and CMH were not provided in the study (main text and figure legends). This problem appears in many places of the manuscript. Please ensure that the points of all results including colonoscopy, stool samples and laboratory examinations should be marked clearly.
2. How many times did every patients enrolled in the study undergo colonoscopy in two years?
3. Please add the primary endpoints and second endpoints in Materials & Methods.

Validity of the findings

1. In table 2 & 3, please add the results of multivariate analyses to find the independent factors which affect the outcomes of mucosal healing and show the results in part of Results.
2. Spearman rank correlation coefficient and P values especially less than 0.05 should be lucidly described.
3. In table 5, some information of the study by Naganuma et al was incomplete.
4. Were the best cutoff values of iFOBT and FC levels reported in others studies? Please discuss them in part of Discussion.

Additional comments

This study focused on the predictive role of stool-based tests, including fecal calprotectin (FC) and the immune fecal occult blood test (iFOBT) for mucosal healing among patients with UC in Taiwan. Multiple researches had demonstrated that the FC and iFOBT had similar roles in predicting (endoscopic mucosal healing) EMH among patients with UC. The present study was the first report from patients with UC in Taiwan that compared iFOBT and FC, which put forward a proposal that given the low cost and similar performance with FC, iFOBT may be advantageous as a first line monitoring tool for patients with UC in Taiwan. This study could guide the clinicians in Taiwan to choose the iFOBT, which is noninvasive and lower cost, as the prective of mucosal healing of UC in clinical practice and decision making.

·

Basic reporting

Overall, the language in the manuscript is clear, concise and acceptable. Introduction is well written and characterizes the context adequately. Minor edits are required on Line 68 (Introduction: add reference on colonoscopy complications.) and Line 96: (Materials & Methods: sentence structure in the inclusion criteria.)

Figures are not yet up to publication standards: Figure 2: legend makes reference to an A and B panel, but only a single panel is available in the document provided. Figure 2 Legend is formatted improperly. In Figures 2-4, no legend is given for the different depicted lines (CIs?).

Raw data is provided, but incompletely defined: Definition and calculation of Mayo score and subscores references four components, scored from 0-3, but partial Mayo scores are said to comprise only three variables; without mention of precisely which three components were selected (Line 103-106: Materials & Methods). Furthermore, raw data includes three mayo scores, without reference to which sub-components they denote; and total Mayo score is not explicitly given in the raw data.

Minor comment: Table 4: Please add asterisks denoting significance levels of the noted spearman correlations.

Experimental design

Yen et al provide a straightforward comparison of fecal calprotectin with the immune fecal occult blood test for the assessment of endoscopic healing. This research bears clinical importance and is of particular relevance in lower resource settings. A declaration with regards to the declaration of Helsinki is missing from the materials and methods.

Methods seem to be described in sufficient detail, though I defer to other reviewers for the quality of the description of the laboratory experiments. Statistical measures are explicitly described and accurately chosen, with the puzzling omission of the construction of a predictive model for the assessment of the predictive value. The authors have elected to impose seemingly arbitrary cutoffs on the predictors.

Validity of the findings

Overall, results are accurate and in-line with expectations. Basic statistical comparisons essentially tell the story and conclusions are in-line with these results. However, the authors stumble in the comparison of the relative predictive value of the FC and iFOBT predictor variables. Cutoff criteria for construction of the ROC are selected without rationale. No predictive model is used, two arbitrary cutoffs are selected (Line 155). This is a textbook example for the use of a logistic regression model. Figures pertaining to the ROC curves lack adequate descriptions and inadequately emphasize the difference between prediction of mucosal healing and complete mucosal healing.

Additional comments

This manuscript asks and answers a pertinent and clinically relevant question and the researchers should be commended for their work in this regard. A need for an alternative option to quantify endoscopic mucosal health exists, and avoiding the high costs associated with FC measurements is preferable. Results of basic statistical analysis support their conclusions, though the assessment and reporting of the relative predictive value of each candidate predictor is not yet up to rigorous scientific standards. If this is solved and figures are brought up to publication standards, I have no doubt this manuscript would be of interest to the scientific and clinical community.

---

## Round 0.2 · accepted · Accept

The manuscript is now ready for publication.

Reviewer 1 ·

Basic reporting

None

Experimental design

None

Validity of the findings

None

Additional comments

Thanks for the authors carefulness and professional. I have no onther confusions and comments on the manuscript.